# Detection and Quantification of Acrylamide in Second Trimester Amniotic Fluid Using a Novel LC-MS/MS Technique to Determine Whether High Acrylamide Content during Pregnancy Is Associated with Fetal Growth

**DOI:** 10.3390/biology12111425

**Published:** 2023-11-13

**Authors:** Nikolaos Vrachnis, Nikolaos Loukas, Nikolaos Antonakopoulos, Niki Maragou, Marios Kostakis, Aliki Tsakni, Dionysios Vrachnis, Despina Vougiouklaki, Nikolaos Machairiotis, Arhodoula Chatzilazarou, Dimitra Houhoula, Rozeta Sokou, Sofoklis Stavros, Peter Drakakis, George Mastorakos, Zoi Iliodromiti

**Affiliations:** 1Third Department of Obstetrics and Gynecology, General University Hospital “Attikon”, Medical School, National and Kapodistrian University of Athens, 12462 Athens, Greeceantonakopoulos2002@yahoo.gr (N.A.); nikolaosmachairiotis@gmail.com (N.M.); sfstavrou@yahoo.com (S.S.); pdrakakis@hotmail.com (P.D.); 2Department of Obstetrics and Gynecology, Tzaneio General Hospital, 18536 Piraeus, Greece; nloux13@hotmail.com; 3Department of Obstetrics and Gynecology, University Hospital of Patras, Medical School, University of Patras, 26500 Patra, Greece; 4Department of Chemistry, National and Kapodistrian University of Athens, Panepistimiopolis Zographou, 15771 Athens, Greece; nmaragkou@uniwa.gr (N.M.); makostak@chem.uoa.gr (M.K.); 5Department of Food Science and Technology, University of West Attica, 12243 Egaleo, Greece; atsakni@uniwa.gr (A.T.); dvougiouklaki@uniwa.gr (D.V.); dhouhoula@uniwa.gr (D.H.); 6Medical School, National and Kapodistrian University of Athens, 11527 Athens, Greece; dionisisvrachnis@gmail.com; 7Department of Wine, Vine and Beverage Sciences, University of West Attica, 12243 Egaleo, Greece; arhchatz@uniwa.gr; 8Neonatal Intensive Care Unit, “Agios Panteleimon” General Hospital of Nikea, 18454 Nikea, Greece; sokourozeta@yahoo.gr; 9Unit of Endocrinology, Diabetes Mellitus and Metabolism, Aretaieio Hospital, Medical School, National and Kapodistrian University of Athens, 11528 Athens, Greece; gmastorak@med.uoa.gr; 10Neonatal Department, Aretaieio Hospital, Medical School, National and Kapodistrian University of Athens, 11526 Athens, Greece

**Keywords:** acrylamide, food contaminant, dietary exposure, liquid chromatography-tandem mass spectrometry, LC-MS/MS, novel technique, validation, amniotic fluid, fetal development, small for gestational age (SGA), fetal growth restriction

## Abstract

**Simple Summary:**

Our study investigated the presence of acrylamide in amniotic fluid and the correlation between maternal exposure to acrylamide and fetal growth. Our amniotic fluid bank included 40 samples from various fetal growth rates, as objectively denoted by the neonatal weight centile at delivery, while our analytical detection method was based on liquid chromatography-tandem mass spectrometry (LC-MS/MS). Acrylamide was determined with reversed phase chromatography and monitoring of two multiple reaction monitoring (MRM) transitions. Quantification was performed using the matrix-matched calibration curve. Acrylamide was detected at concentrations between 7.1 and 1468 ng/mL in six out of the total of 40 amniotic fluid samples. Detection of acrylamide in early second trimester amniotic fluid raises concerns about fetal health, given that published data on animal studies have attributed a number of birth defects to acrylamide. Our novel LC-MS/MS method for the determination of acrylamide in amniotic fluid proved to be effective and its performance in practice was very accurate, simple, and fast.

**Abstract:**

Introduction: Acrylamide, an organic compound, is, chemically speaking, a vinyl-substituted primary amide. It is produced industrially, principally as a precursor to polyacrylamides, for use in such products as plastics and cosmetics. This same compound, however, forms naturally in certain foods, both home-cooked and packaged, especially when prepared at high temperatures. We developed and validated a novel reliable technique for the determination of acrylamide in amniotic fluid. Multiple reaction monitoring (MRM) is a targeted mass spectrometry (MS) technique which enables the detection and quantification of particular molecules in a complex mixture. Thanks to its throughput, selectivity, and sensitivity, MRM-MS has been identified as offering an alternative to antibody-based studies for the purpose of biomarker verification. Our aim was to investigate the presence of acrylamide in amniotic fluid and, via the MRM-MS technique, to determine whether there is any correlation between maternal exposure to acrylamide, through a woman’s diet, and fetal growth. Methods: Our amniotic fluid bank included 40 samples from various fetal growth rates, as objectively denoted by the neonatal weight centile at delivery, while our analytical detection method was based on liquid chromatography-tandem mass spectrometry (LC-MS/MS). Acrylamide was determined with reversed phase chromatography and monitoring of two multiple reaction monitoring (MRM) transitions. Quantification was performed using the matrix-matched calibration curve. Results: Acrylamide was detected at concentrations between 7.1 and 1468 ng/mL in six out of the total of 40 amniotic fluid samples that were used. Our method limit of detection and quantification was 1.4 ng/mL and 4.6 ng/mL, respectively. The repeatability of our method ranged between 11 and 14%, expressed as relative standard deviation levels between 5 and 100 ng/mL. Conclusions: Detection of acrylamide in early second trimester amniotic fluid, for the first time in the literature to our knowledge, raises concerns about fetal health, given that published data on animal studies have attributed a number of birth defects to acrylamide. Our novel LC-MS/MS method for the determination of acrylamide in amniotic fluid proved to be effective and its performance in practice was very accurate, simple, and fast. Validation of the method revealed that the use of a matrix-matched curve is necessary for the quantification.

## 1. Introduction

Acrylamide is a white solid organic compound which is odorless and soluble in water and in a number of organic solvents. A vinyl-substituted primary amide, it is produced industrially, chiefly as a precursor to polyacrylamides, with the latter being used as water-soluble thickeners and flocculation agents mainly for industrial processes. However, acrylamide also forms naturally in the burnt areas of starchy foods, that is, as a result of high-temperature cooking (i.e., frying, roasting, and baking) [1,2], being found in such products as potatoes [3,4], bread [4,5], and coffee [4], as well as in cosmetic products [6]. While there has been intensive investigation into the presence of acrylamide in numerous food samples, this has not been the case concerning biological samples [7,8].

Despite the fears concerning its effects on people’s health subsequent to its discovery in 2002, the general opinion today is that dietary acrylamide is not carcinogenic to humans, while the American and British Cancer Societies rejected the proposal that burnt food can cause cancer as a “myth” [9,10]. On the other hand, acrylamide is still considered to be neurotoxic for humans and animals. A dose-dependent weight reduction and skeletal malformations have been reported in rodent offspring exposed to acrylamide in utero [11,12,13]. Acrylamide and its metabolites are also known to form adducts with proteins and DNA, raising concerns for genotoxicity [14]. Moreover, evidence of the adverse developmental effects of acrylamides on ovarian function and fertility spanning at least two consecutive generations underlines the necessity for more targeted strategies during pregnancy, including following a healthy diet and eliminating consumption of acrylamide-rich products [15].

Acrylamide is a small highly water-soluble organic molecule with low molecular weight (MW: 71 g/mol), thus rendering its analytical determination a challenge. The analytical technique commonly applied is liquid chromatography coupled to mass spectrometry. Chromatographic separation has been reported to be achieved with typical reversed phase analytical columns C18 [4,6], graphitized carbon columns [7], and a hydrophilic interaction chromatography column (HILIC) [5].

In our study, we aimed to determine whether acrylamide is present in human amniotic fluid, which, if confirmed, would reflect its presence in the fetus. This is the first time, as far as we know, that such an attempt has been made to detect and quantify this organic compound in the mother and fetus. For this specific purpose we developed a novel quantification technique for acrylamide, based on liquid chromatography-tandem mass spectrometry (LC-ESI-MS/MS), which we present herein.

## 2. Materials and Methods

### 2.1. Amniotic Fluid Samples

#### 2.1.1. Amniotic Fluid Collection

Amniotic fluid samples were collected from 40 women who had undergone amniocentesis early in the 2nd trimester of gestation (15–22 weeks) based on various indications, such as advanced maternal age, previous history of birth defects, increased nuchal translucency, or detection of an anomaly in the ultrasound examination of the first or second trimester. Twin pregnancies and current pregnancies with fetuses of abnormal karyotype or with severe congenital malformations were excluded from the study.

All women participating in the study provided informed consent. The study was approved by the Scientific Committee of Attikon General University Hospital, Athens, Greece.

#### 2.1.2. Amniotic Fluid Samples Preparation and Storage

Immediately following amniocentesis, the amniotic fluid samples were centrifuged and the supernatants were stored in polypropylene tubes at −80 °C. All pregnancies were followed up until delivery and the corresponding amniotic fluid samples were then withdrawn from our amniotic fluid sample bank and acrylamide was measured in order to study its levels and compare those of normal growth pregnancies at term (control group) with those of the groups of embryos of at term pregnancies with residual and enhanced growth.

#### 2.1.3. Determination of Study Groups

After fetal growth patterns and birth weights had been recorded, all 40 samples were divided into three groups, namely, SGA (small for gestational age), AGA (appropriate for gestational age), and LGA (large for gestational age). To allocate the centile of each neonate at delivery, a gestation-related weight computer program was applied [16]. Our study sample was composed of two SGA fetuses and three LGA fetuses, matched for gestational age, sex, maternal height and weight, and 35 AGA fetuses.

### 2.2. Novel Quantification Technique for Acrylamide Based on Liquid Chromatography-Tandem Mass Spectrometry

#### 2.2.1. Chemicals and Reagents

We used acrylamide standard (100%) from Ehrenstorfer (Augsburg, Germany). A stock solution (100 μg/mL) was prepared in methanol, used for further dilutions, and stored at −15 °C. Methanol (MeOH) and acetonitrile (ACN) LC-MS grade were acquired from Merck (Darmstadt, Germany). Ammonium formate and formic acid (LC-MS grade) were obtained from Fluka (Buchs, Switzerland). Water used as HPLC solvent, and for the sample preparation it was purified with a Milli-Q water system (Millipore, Bedford, MA, USA). Regenerated cellulose syringe filters (RC) of 15 mm diameter and 0.2 μm pore size were obtained from Phenomenex (Torrance, CA, USA).

#### 2.2.2. Sample Preparation

A volume of 200 μL of amniotic liquid was diluted with 200 μL of MeOH; the solution was vortexed and filtered with regenerated cellulose syringe filters with a pore size of 0.2 μm before measurement with LC–MS/MS. For samples with available volume below 200 μL or with detected concentration outside the linear range, an extra dilution was performed with water/methanol 1/1, *v*/*v*.

#### 2.2.3. LC-MS/MS Measurements

##### LC-MS/MS Optimization

Liquid chromatography-tandem mass spectrometry (LC-MS/MS) was applied for the identification and quantification of acrylamide in amniotic liquid samples using an AB SCIEX QTRAP ready 5500+ triple quadrupole mass analyzer with a Turbo V electrospray ionization interface (ESI) connected to a SCIEX EXION LC AD system consisting of the binary gradient pump, the autosampler, the column oven, and the system controller. Data acquisition and processing were performed with Analyst version 1.7 and SCIEX OS version 2.1.6.59781 software, respectively.

The selection of the LC-MS/MS conditions was based on previously published methods for the determination of acrylamide in food samples [4,5,11] or in-house optimization experiments. In more detail, the first step was the selection of the multiple reaction monitoring (MRM) optimum parameters, which was performed with infusion experiments of 100 ng/mL acrylamide standard solution in MeOH/water (50/50). Afterwards, preliminary liquid chromatography experiments with two different reversed phase analytical columns and one hydrophilic interaction chromatography column (HILIC) were performed with standard solutions of acrylamide, a pooled sample of amniotic fluid, and a spiked pooled sample of amniotic fluid. The tested analytical columns were as follows: (i) a Kinetex XB-C18 (100 mm × 2.1 mm, 2.6 μm); (ii) a Phenomenex Luna C18 analytical column (150 mm × 4.6 mm, 5 μm) with mobile phase consisting of 0.1% *v*/*v* formic acid in water (solvent A) and methanol (solvent B) at a ratio of 90/10 (A/B); and (iii) a TSKgel Amide-80 (250 mm × 4.6 mm, 5 μm) with mobile phase consisting of 1 mM ammonium formate and 0.1% (*v*/*v*) formic acid in water (solvent A) and acetonitrile (solvent B) at ratio 5/95 (A/B). After the selection of the chromatographic conditions, the electrospray ion source parameters were finalized according to the manufacturer’s instructions based on the selected mobile phase and its flow rate.

##### LC-MS/MS Conditions

The final optimum LC-MS/MS conditions were as follows: chromatographic separation was performed with a Phenomenex Luna C18 analytical column (150 mm × 4.6 mm) with a particle size of 5 μm and a pore diameter of 100 Å under isocratic elution at 0.5 mL/min flow rate, for 8 min run time. The mobile phase consisted of 0.1% *v*/*v* formic acid in water (solvent A) and methanol (solvent B) at a ratio of 90/10 (A/B); the column oven temperature was set at 25 °C and the injection volume at 10 μL.

The ion source spray voltage was set at 5.5 kV and the temperature at 400 °C. The curtain gas, which protects the ion entrance optics from ambient air and solvent droplets contamination, was set at 20 psi, while the nebulizer gas (GS1) and the heater gas (GS2) were set at 40 and 50 psi, respectively. For multiple reaction monitoring (MRM), the *m*/*z* ion of the protonated parent compound [M+H]+ was found to be 72.0 Da with an optimum declustering potential of 71 V, and the *m*/*z* of the two most intense product ions were 55.0 Da and 27.2 Da. The collision gas was set at 8 psi, and the collision energy for the quantification transition (72.0 > 55.0) and the confirmation transition (72.0 > 27.2) were 17 V and 31 V, respectively. Accordingly, the collision cell exit potential, which focuses and accelerates the product ions out of the collision cell (Q2) and into the filtering quadrupole Q3, were 8 and 12 V for the quantification and confirmation transitions, respectively. The dwell time for each transition was set at 150 ms.

#### 2.2.4. Method Validation

The developed method was evaluated using standard solutions of acrylamide prepared in methanol/water (1/1, *v*/*v*) and a pool sample of amniotic fluid prepared by mixing 50 or 100 μL from samples S1-S40 and spiked with known amounts of acrylamide.

The linearity of the response of the LC-ESI-MS/MS system versus acrylamide concentration was examined with a standard calibration curve constructed by measuring standard solutions of 1, 10, 50, and 100 ng/mL acrylamide concentrations. A matrix-matched calibration curve was prepared with the pool sample spiked with the target analyte at four levels between 5 and 100 ng/mL. The concluding linear equation of the matrix-matched calibration curve was derived following the subtraction of the signal of the unfortified sample from the signal of the fortified samples. The method limit of detection (LOD) was defined as the concentration of acrylamide in matrix with a peak area corresponding to three times the average level of the baseline noise close to the peak (S/N = 3). The method limit of quantification (LOQ) was defined as the concentration of acrylamide in the matrix with a peak area corresponding to ten times the average level of the baseline noise close to the peak (S/N = 10).

In order to assess precision and accuracy, the method was applied to the pool sample, which was spiked with acrylamide at four fortification levels, namely, 5, 10, 50, and 100 ng/mL, and analyzed in multiple replicates. The recovery (%R) of the method was calculated by subtracting the concentration measured in the non-spiked sample from the concentration measured in the spiked sample, then dividing it with the spiked concentration ($C_{ADDED}$) according to the following equation:$${\%R}_{}\;=\;\;\frac{C_{SPIKED\;SAMPLE}-C_{NONSPIKED\;SAMPLE}}{C_{ADDED}}\times 100$$

## 3. Results

### 3.1. Novel Method Performance

Figure 1 and Figure 2 illustrate the quantification and confirmation transitions MRM, (Q) and (C), respectively, acquired for the solvent MeOH/water (1/1, *v*/*v*), the standard solution of 10 ng/mL, the pool sample of amniotic fluid which contained 4.4 ng/mL acrylamide, and the same pool sample fortified at 5 ng/mL.

Table 1 depicts the equations of the calibration curves using spiked amniotic fluid samples as calibration standards. The standard deviation of the slope and the intercept and the correlation coefficient of each equation are also given. It is demonstrated that the method presented linearity for the tested concentration range with correlation coefficients exceeding 0.99. Also displayed in Table 1 are the LODs and LOQs of the method. Comparison of the slopes of the matrix calibration curve with the standard calibration curve of Table 1 reveals a positive matrix effect, which is also reflected in the recoveries, presented in Table 2, for all fortification levels which are above 100%. Overestimation of acrylamide in coffee samples has been reported previously due to close elution of interfering compounds, such as N-Acetyl-β-alanine, 3-aminopropanamide, and lactamide, and a similar in-source fragmentation pattern [11]. In the present study, the quantification of the acrylamide detected in the samples was performed with a matrix-matched calibration curve in order to avoid overestimation of acrylamide content and overcome the existence of the matrix effect. The precision data expressed as relative standard deviation (% RSD) are also summarized in Table 2.

### 3.2. Detection of Acrylamide

The optimized method was applied to 40 samples of amniotic fluid. Acrylamide was detected in six samples at concentration levels between 7.1 and 1468 ng/mL (Table 3).

## 4. Discussion—Clinical Implications

Given the almost inevitable maternal exposure to acrylamide and other potentially harmful chemicals, the present study aimed to confirm fetal exposure to acrylamide from early second trimester gestational age, which is the sensitive phase of fetal organogenesis of most fetal organs having been completed a few weeks earlier, with the exception of the fetal brain, whose development continues throughout pregnancy [14,16]. Firm evidence of this early exposure has been verified by the detection of acrylamide in amniotic fluid samples retrieved in the early second trimester of pregnancy. Although acrylamide’s presence has been intensively investigated in a variety of food samples, reports on biological samples are lacking. Several factors have been investigated for their role in pregnancy complications [17,18].

It has been shown in a recent systematic review and meta-analysis of cohort studies that there is an association between maternal high acrylamide exposure and significantly lower birth weight of offspring. An association was also observed between exposure to maternal acrylamide and small for gestational age neonates (OR 1.14, *p* < 0.001). Furthermore, pooled ORs indicated that fetuses exposed to acrylamide were at high risk of developing obesity later in life, pointing to a possible effect of early metabolic programming of these children starting from their fetal period [19]. The purpose of our study was to evaluate fetal exposure more objectively by studying the amniotic fluid and not based on maternal nutritional habits. Amniotic fluid measurements do not necessarily relate to maternal and placental exposure to acrylamide, as the placental barrier may work differently at this early stage of pregnancy, and the acrylamide may or may not pass to the fetus, despite the maternal exposure. Furthermore, it is extremely difficult to accurately determine maternal exposure to acrylamide with the use of questionnaires, as the same source of food may be cooked for a longer or shorter time, in different ways, etc., with these parameters affecting the acrylamide levels in the pregnant. Finally, maternal and fetal exposure are expected to be comparable but not the same occurrences as several factors are known to affect the amount of acrylamide that ultimately reaches to the fetus, such as the placental barrier explained above or the maternal toxic substance clearance mechanisms.

This is the first time, to our knowledge, that acrylamide has been detected in amniotic fluid samples. Given the harmful effects of acrylamide, our study offers unique and valuable information on the possible impact of mothers’ dietary exposure to acrylamide sources on their fetuses starting from the very sensitive phase of intrauterine organogenesis. Furthermore, we divided our samples into different fetal growth velocity groups in order to determine whether there is a direct correlation between the level of fetal exposure to acrylamide and fetal growth disturbances, such as impaired or augmented growth.

The fact that acrylamide is a small highly water-soluble organic molecule with a low molecular weight of 71 g/mol renders its analytical determination a challenge. The analytical technique commonly applied is liquid chromatography coupled to mass spectrometry. However, due to the restrictions pertaining to common quantification methods and the difficulty of obtaining sufficient amniotic fluid volumes, we developed the concept of a new quantification technique for acrylamide determination in amniotic fluid, based on liquid chromatography-tandem mass spectrometry (LC-ESI-MS/MS), as described above. This technique proved to be relatively cheap, simple, fast, sensitive, and reliable. The validation of the method revealed that there is a signal enhancement in samples of amniotic fluid and, therefore, the use of the matrix-matched curve is necessary for the quantification. The method was successfully applied to 40 samples, and acrylamide was detected in 15% of the samples.

Our samples came from 40 amniocentesis cases of various fetal growth centiles (as shown in Table 3). Our aim was to investigate the correlation between maternal exposure to acrylamide and fetal growth, comparing three groups, based on fetal growth potential, namely, small for gestational age (SGA) fetuses, large for gestational age (LGA) fetuses, and appropriate for gestational age (AGA) fetuses (control). Acrylamide concentration in amniotic fluid is considered to be representative of direct fetal blood exposure through the maternal-fetal blood exchange in the placental bed. Second trimester amniotic fluid is known to accurately reflect fetal circulation and this is the reason that, as an example, amniotic fluid samples are used in order to objectively evaluate fetal exposure to harmful factors after maternal exposure, such as cases of congenital infections, when fetal exposure (infection) needs to be investigated [20]. Even more importantly, the earlier the week of pregnancy the more vulnerable a fetus can be when exposed to potentially harmful factors. Thus, identifying acrylamide in the early second trimester is particularly crucial, given that the fetus is expected to be at a very vulnerable stage of growth.

Interestingly, acrylamide was not detected in most of our samples. Only six out of 40 amniotic fluid samples returned quantifiable results. The most possible explanation for this could be the presence of acrylamide concentrations below our limit of detection (1.4 ng/mL). Another explanation might be the rapid degradation of the substance. Finally, different levels of maternal exposure are expected, reflecting the different maternal dietary profiles along with the different rates of acrylamide degradation in the maternal circulation. This is consistent with the wide range of concentrations detected in our samples (7.1–1468 ng/mL).

Acrylamide is metabolized to glycidamide via the CYP2E1 enzyme, and both are excreted through the urine as N-Acetyl-S-(2-carbamoylethyl)-L-cysteine (AAMA) and N-Acetyl-S-(2-carbamoyl-2-hydroxyethyl)-L-cysteine (GAMA) after conjugation with glutathione via the glutathione S-transferase. Previous studies in the literature have included blood samples in which acrylamide and glycidamide hemoglobin adducts were measured. Ninety percent of acrylamide intake is excreted through the urine mainly as AAMA.

Nevertheless, acrylamide was detected in six samples, all of them in the AGA group. This is clear evidence that acrylamide crosses the placenta into the fetal circulation and is secreted in the fetal urine. Acrylamide has for quite long been known to be a neurotoxin affecting both humans and animals, while, in the past, it was also classified as possibly being a human carcinogen (International Agency for Research on Cancer, IARC 1994), though the current opinion as to its carcinogenic potential is very different, as previously mentioned. Prenatal exposure to acrylamide is of particular concern due to its proven developmental toxicity, mainly reported in rodent studies, dose-dependent weight reduction, and skeletal malformations, having been reported in rodent offspring exposed to acrylamide in utero [21,22,23]. Acrylamide and glycidamide may form adducts with proteins and DNA, thus raising concerns about genotoxicity [24].

Data from the European Prospective Mother-Child Study (NewGeneris) [25] have provided strong evidence that high dietary exposure to acrylamide during pregnancy is linked to reduced birth weight and head circumference. Interestingly, birth weight decreased in proportion to increased acrylamide cord blood level. The implication of this finding is significant, as head circumference is an important indicator of continuous brain growth, while reduced birth head circumference has been associated with impaired neurodevelopment [26]. As reported in another cord blood study, prenatal dietary acrylamide exposure in the form of its metabolite glycidamide has been found to be inversely associated with birth weight as well as neonatal length and head circumference [27]. Acrylamide’s ability to react with proteins, including enzymes, receptors, and cytoskeletal proteins means that it may affect a wide array of intra- and intercellular processes, which is likely to account for the toxic biological actions it exerts [26].

In our study, acrylamide was detected in only six out of 35 AGA cases. The short-term outcome for all neonates included in our study was normal. This period reflects the time of birth up until discharge of the neonate from the hospital. Longer-term consequences were not investigated in our study; they are considered to be a reflection of multiple other parameters as well. However, it would be of interest for future studies to focus on this period of neonatal and infant life.

Based on our results, we could not identify any correlation between acrylamide concentrations and fetal growth centiles. This was due to the small number of abnormal growth velocity samples, with the small number of cases included in our study being a limitation of our work. Additional factors which influence fetal growth, such as placental invasion and function and gestational diabetes, have a more crucial effect than environmental factors or nutrition. On the other hand, acrylamide may simply be a marker of a less healthy diet in general, and its negative effect on fetal growth may thus be partially attributable to this rather than to acrylamide directly. Nonetheless, given the evidence in the literature that maternal intake of foods with high acrylamide content during pregnancy has been associated with subnormal head circumference and birth weight, it is clear that large multicenter studies are needed to elucidate the exact role of acrylamide in fetal growth disturbances from the early second trimester of pregnancy.

## 5. Conclusions

This is the first time, to our knowledge, that acrylamide has been detected in amniotic fluid samples. This is also the first time that our new quantification method has been used. The development and performance of this novel LC-ESI(+)-MS/MS triple quadrupole method for the determination of acrylamide in amniotic fluid is herein presented. The method was successfully applied to 40 samples, and acrylamide was detected in 15% of the tested samples. The validation of the method revealed that there is a signal enhancement in samples of amniotic fluid and, therefore, the use of the matrix-matched curve is essential for the quantification. Acrylamide detection in amniotic fluid raises concerns about fetal health given the literature data confirming its harmful effects, to date mainly in animal studies. Further larger studies additionally focusing on acrylamide metabolites are required to determine the exact effect of acrylamide on fetal growth and the factors that influence fetal exposure and vulnerability in order to establish preventive policies and advice about prophylactic measures in pregnant women.

## Figures and Tables

**Figure 1 biology-12-01425-f001:**
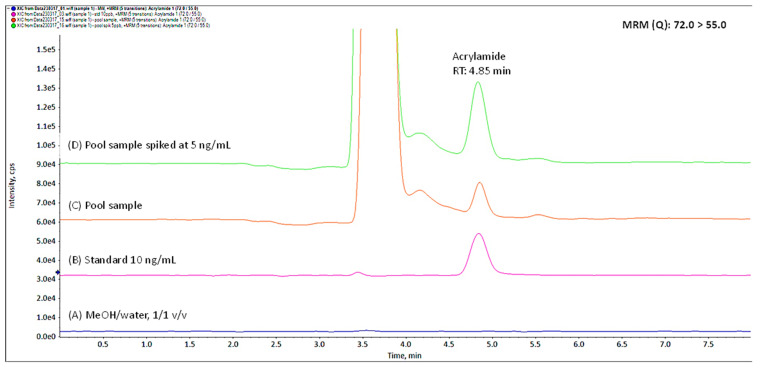
Quantification MRM (72 > 55) of (**A**) solvent, (**B**) standard solution 10 ng/mL, (**C**) unspiked pool sample containing 4.4 ng/mL, and (**D**) pool sample spiked at 5 ng/mL.

**Figure 2 biology-12-01425-f002:**
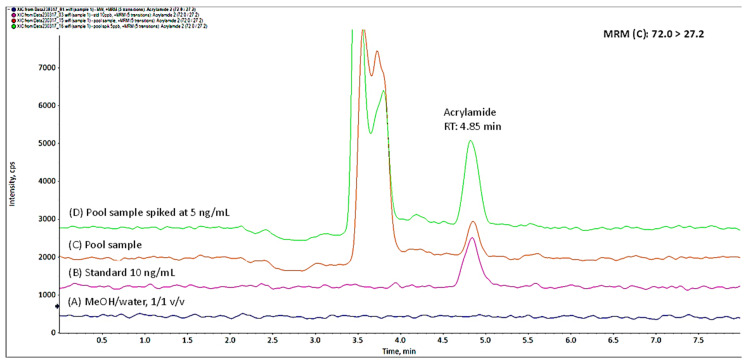
Confirmation MRM (72 > 27.2) of (**A**) solvent, (**B**) standard solution 10 ng/mL, (**C**) unspiked pool sample containing 4.4 ng/mL, and (**D**) pool sample spiked at 5 ng/mL.

**Table 1 biology-12-01425-t001:** Calibration curves, LOD (limit of detection), and LOQ (limit of quantification) for standard solution and for amniotic fluid. C: ng/mL. Concentration range: 1–100 ng/mL for standard solutions and 5–100 ng/mL for amniotic fluid.

Matrix	Calibration Curve (× 10^−4^)	LOD (ng/mL)	LOQ (ng/mL)
Standard solutions	y = (2.70 ± 0.09) × C + (7.95 ± 4.66) r = 0.998	0.7	2.3
Amniotic fluid	y = (3.33 ± 0.05) × C + (12.18 ± 2.34) r = 0.993	1.4	4.6

**Table 2 biology-12-01425-t002:** Recovery and repeatability data for spiked amniotic fluid samples at four different levels (RSD = relative standard deviation).

Fortification Level (ng/mL) (Number of Replicates)	Average Recovery (%)	%RSD
5 ppb (n = 4)	127	14
10 ppb (n = 7)	144	12
50 ppb (n = 3)	132	11
100 ppb (n = 3)	136	14

**Table 3 biology-12-01425-t003:** Acrylamide concentrations in amniotic fluid samples. Forty samples were analyzed in total and acrylamide was detected in six samples (LOD = 1.4 ng/mL) (15% detection rate).

Sample Number	Fetal Birth Weight Centile	Acrylamide Concentration (ng/mL)
6	18	1468
7	20	71
15	36	7.6
16	38	47.8
20	49	8.1
33	79	122
1–5, 8–14, 17–19, 21–32, 34–40	Various	Not detected

## Data Availability

Data available on reasonable request.

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
