# Peer review of "Detection and Quantification of Acrylamide in Second Trimester Amniotic Fluid Using a Novel LC-MS/MS Technique to Determine Whether High Acrylamide Content during Pregnancy Is Associated with Fetal Growth"

_biology, 2023, doi:10.3390/biology12111425_

Round 1
Reviewer 1 Report
Comments and Suggestions for Authors
Detection and quantification of acrylamide in second trimester 2 amniotic fluid using a novel LC-MS/MS technique to determine 3 whether high acrylamide content during pregnancy is associated with fetal growth
Nikolaos Vrachnis 1# , Nikolaos Loukas 2#, Nikolaos Antonakopoulos 1,3, Niki Maragou 4,5, Marios Kostakis 4 , Aliki 6 Tsakni 5 , Dionysios Vrachnis 7 , Despina Vougiouklaki 5 , Nikolaos Machairiotis 1 , Archodoula Chatzilazarou 6 , Di- 7 mitra Houhoula 5 , Rozeta Sokou 8 , Sofoklis Stavros 1 , Petros Drakakis 1 , George Mastorakos 9 and Z. Iliodromiti
The paper includes a detailed description of the establishment and validation of a method for measuring acrylamide in amniotic fluid. The method is well described and referenced The results on 40 amniotic fluid samples collected in the second trimester of gestation (15-22 weeks) are given . Only 6 specimens had a measureable level of acrylamide that were above the limit of detection of 1.4ng/ml and these were found in the appropriate for gestational age group.
There are a few English corrections that are suggested:
Line 40; please describe the words “growth centiles” How are the growth centiles determined?
Line 47 Delete the words “for fortification”
The repeatability of our method ranged between 11 and 14%, expressed as relative standard deviation for fortification levels between 5 and 100 ng/mL.
Line 77; Suggest replacing “this” with “thus”
Line 101; Suggest replacing “while” with “and”
Line 142; Suggest replacing “on” with “or”
Lines 313 to 316. And lines 318 to 320. Please revise the wording as the reviewer did not completely understand.
1. What is the main question addressed by the research? The aim was to determine if there is any correlation between maternal exposure to acrylamide and fetal growth by measurement of acrylamide in early second trimester amniotic fluid. And comparing the levels of acrylamide with the size of the infant at birth.
The paper includes a detailed description of the establishment and validation of a method for measuring acrylamide in amniotic fluid. The method is well described and referenced The results on 40 amniotic fluid samples collected in the second trimester of gestation (15-22 weeks) are given . Only 6 specimens had a measureable level of acrylamide that were above the limit of detection of 1.4ng/ml and these were found in the appropriate for gestational age group.
2. Do you consider the topic original or relevant in the field? Does it address a specific gap in the field? The authors gave recent referenced reports of the presence and measurement of acrylamide in foods. The key reference that prompted the development of a method to measure acrylamide in amniotic fluid, was the 2021 article, references #12 and # 13 that describe the association of increased Bisphenol A, phthalates, including acrylamide, and their possible association with fetal growth. However, the Bisphenol A and phthalates are endocrine disruptors, and acrylamide has a different mode of toxicity. ( See the Hogervorst article listed below.) In addition there were references to birth defects in rodents that were associated with birth defects. As there was no report of measurement of acrylamide in amniotic fluid in the literature, there was a need to develop a sensitive and reliable test.
3. What does it add to the subject area compared with other published material? The article describes a sensitive and reproducible method for the measurement of acrylamide in amniotic fluid.
4. What specific improvements should the authors consider regarding the methodology? What further controls should be considered? The authors should consider extending their study to assay acrylamide in cord blood samples collected at birth.
5. Are the conclusions consistent with the evidence and arguments presented? Yes
6. Are the references appropriate? The following recent references are not in this article and recommend that they be added. Cantrell MS, McDougal OM. Biomedical rationale for acrylamide regulation and methods of detection. Compr Rev Food Sci Food Saf. 2021 Mar;20(2):2176-2205. doi: 10.1111/1541-4337.12696. Epub 2021 Jan 23. PMID: 33484492; PMCID: PMC8394876. Hogervorst J, Vesper HW, Madhloum N, Gyselaers W, Nawrot T. Cord blood acrylamide levels and birth size, and interactions with genetic variants in acrylamide-metabolising genes. Environ Health. 2021;20(1):35. Published 2021 Apr 1. doi:10.1186/s12940-021-00715-0 Zhan Y, Xiao Y, Guan T, Zhang S, Jiang Y. Relationship between gestational acrylamide exposure and offspring's growth: a systematic review and meta-analysis of cohort studies. Public Health Nutr. 2020;23(10):1791-1799. doi:10.1017/S1368980019005123 Aldawood N, Jalouli M, Alrezaki A, et al. Fetal programming: in utero exposure to acrylamide leads to intergenerational disrupted ovarian function and accelerated ovarian aging. Aging (Albany NY). 2022;14(17):6887-6904. doi:10.18632/aging.204269
7. Please include any additional comments on the tables and figures. Tables 1 and 2 are part of the method development and are useful for implementing the analysis of acrylamide by LC-MS/MS. Table 3 could be condensed to report only those amniotic fluid samples that have detectable levels of acrylamide
Comments on the Quality of English Language
-
Author Response
Dear Reviewer 1,
Thank you for your constructive comments. We have addressed all your comments. I understand that I will upload my revised manuscript after your review.
Below there follows a point-by-point response to your comments:
- There are a few English corrections that are suggested:
Line 40; please describe the words “growth centiles” How are the growth centiles determined?
Thank you for your comment. The word “centiles” has been changed to the word “rates” in the Abstract. This part of the Abstract now reads as follows:
“Our amniotic fluid bank included 40 samples from various fetal growth rates, as objectively denoted by the neonatal weight centile at delivery, while our analytical detection method is based on liquid chromatography-tandem mass spectrometry (LC-MS/MS).”
Because this is in the Abstract we cannot use many words to explain it. However, in the main text, in the methods section that follows, the term growth centile is explained and the formula of calculation we used is provided within the respective reference. The text there reads:
“After fetal growth patterns and birth weights had been recorded, all 40 samples were divided into three groups, namely, SGA (small for gestational age), AGA (appropriate for gestational age), and LGA (large for gestational age). To allocate the centile of each neonate at delivery, a gestation-related weight computer program was applied [16].”
- Line 47 Delete the words “for fortification”
The repeatability of our method ranged between 11 and 14%, expressed as relative standard deviation for fortification levels between 5 and 100 ng/mL.
Thank you for your comment. Both words were deleted.
- Line 77; Suggest replacing “this” with “thus”
Thank you for your comment. A replacement has now been made.
- Line 101; Suggest replacing “while” with “and”
Thank you for your comment. This is now done.
- Line 142; Suggest replacing “on” with “or”
Thank you for your comment. Replaced.
- Lines 313 to 316. And lines 318 to 320. Please revise the wording as the reviewer did not completely understand.
Thank you for your comment. The sentence previously read: “It is common practice, in cases such as congenital infections when fetal exposure (infection) needs to be investigated, for amniotic fluid to be sampled and examined in the second trimester since it accurately reflects fetal circulation [20].”.
Now, after your advice it reads:
“Second trimester amniotic fluid is known to accurately reflect fetal circulation and this is the reason that, as an example, amniotic fluid samples are used in order to objectively evaluate fetal exposure to harmful factors after maternal exposure, such as cases of congenital infections, when fetal exposure (infection) needs to be investigated [20].”.
This paragraph gives the conceptualization of our study material, which concerns second trimester amniotic fluid.
- The authors should consider extending their study to assay acrylamide in cord blood samples collected at birth.
Thank you for your comment. Indeed, acrylamide measurement in cord blood samples is also of great interest. This can be part of a future project. However, identifying this potentially harmful molecule in the early second trimester is even more crucial for a fetus as it is in a more vulnerable period. The earlier the week of pregnancy the more vulnerable a fetus can be. A cord blood sample is collected at the time of birth. That is much later: in the late second or third trimester. A comment about this was included in our manuscript:
“Even more importantly, the earlier the week of pregnancy the more vulnerable a fetus can be when exposed to potentially harmful factors. Thus, identifying acrylamide in the early second trimester is particularly crucial, given that the fetus is expected to be at a very vulnerable stage of growth.”
- The following recent references are not in this article and recommend that they be added. Cantrell MS, McDougal OM. Biomedical rationale for acrylamide regulation and methods of detection. Compr Rev Food Sci Food Saf. 2021 Mar;20(2):2176-2205. doi: 10.1111/1541-4337.12696. Epub 2021 Jan 23. PMID: 33484492; PMCID: PMC8394876.Hogervorst J, Vesper HW, Madhloum N, Gyselaers W, Nawrot T. Cord blood acrylamide levels and birth size, and interactions with genetic variants in acrylamide-metabolising genes. Environ Health. 2021;20(1):35. Published 2021 Apr 1. doi:10.1186/s12940-021-00715-0 Zhan Y, Xiao Y, Guan T, Zhang S, Jiang Y. Relationship between gestational acrylamide exposure and offspring's growth: a systematic review and meta-analysis of cohort studies. Public Health Nutr. 2020;23(10):1791-1799. doi:10.1017/S1368980019005123 Aldawood N, Jalouli M, Alrezaki A, et al. Fetal programming: in uteroexposure to acrylamide leads to intergenerational disrupted ovarian function and accelerated ovarian aging. Aging (Albany NY). 2022;14(17):6887-6904. doi:10.18632/aging.204269
Thank you for your comment. The proposed recent literature was added to the manuscript. Furthermore, new text related to each one of the above references was added to the manuscript, as follows:
The reference of Cantrell et al. was added as a reference with the following text:
“While there has been intensive investigation into the presence of acrylamide in numerous food samples, this has not been the case as concerns biological samples [7,8].”
For the reference of Hogervorst et al. we added:
“As reported in a cord blood study, prenatal dietary acrylamide exposure in the form of its metabolite glycidamide has been found to be inversely associated with birth weight as well as length and head circumference [27].”
For the reference of Zhan et al. we added:
“It has been shown in a recent systematic review and meta-analysis of cohort studies that there is an association between maternal high acrylamide exposure and significantly lower birth weight of offspring. An association was also observed between exposure to maternal acrylamide and small for gestational age neonates (OR 1•14, P < 0•001). Furthermore, pooled ORs indicated that fetuses exposed to acrylamide were at high risk of developing obesity later in life, this pointing to a possible effect of early metabolic programming of these children starting from their fetal period [19].”
For the reference of Aldawood et al. we added:
“Evidence of the adverse developmental effects of acrylamides on ovarian function and fertility spanning at least two consecutive generations underlines the necessity for more targeted strategies during pregnancy, including following a healthy diet and eliminating consumption of acrylamide-rich products [15].”
- Tables 1 and 2 are part of the method development and are useful for implementing the analysis of acrylamide by LC-MS/MS. Table 3 could be condensed to report only those amniotic fluid samples that have detectable levels of acrylamide.
Thank you for your comment. Table 3 has now been condensed so as to report only the amniotic fluid samples that have detectable levels of acrylamide.
Reviewer 2 Report
Comments and Suggestions for Authors
Vrachnis and colleagues in this paper wanted to determinate the presence and the quantity of acrylamide in amniotic fluid. the study is interesting and have a good scientific sound but in my opinion the presence of acrylamide in their samples should be better analyze. For examples what could be the correlation with the women in which you find it in amniotic fluid? Do you know something about their alimentation? Or...In the sample in which you find acrylamide, do you have some notice about the baby? Are they in health, what about their delivery? These informations could bring us to postulate a role of this test like very important to diagnostic test in correlation with amniocentesis, or could bring us to a gestational alimentation advice with respect to another.
Please, could give me some information about it or if you haven't, discuss it?
Author Response
Dear Reviewer 2,
Thank you for your constructive comments. All manuscript issues that you discussed have now been addressed. Please find below a point-by-point response to your comments:
- Vrachnis and colleagues in this paper wanted to determinate the presence and the quantity of acrylamide in amniotic fluid. the study is interesting and have a good scientific sound but in my opinion the presence of acrylamide in their samples should be better analyze. For examples what could be the correlation with the women in which you find it in amniotic fluid? Do you know something about their alimentation?
Thank you for your comment. The purpose of our study was not to examine fetal exposure based on maternal nutritional habits, but, for the first time, to provide evidence of any fetal exposure and existence of acrylamide by studying it in the amniotic fluid. Our aims were multiple: to detect it, prove that our novel method works, measure its concentration, and, if possible, to correlate its concentrations with fetal growth. On the other hand, amniotic fluid measurements do not necessarily relate to maternal and placental exposure to acrylamide, as it may not pass to the fetus despite maternal exposure. Moreover, it is difficult to determine accurately maternal exposure to acrylamide with the use of questionnaires, as the same source of food may be cooked for a longer or shorter time, in different ways, etc. which affects acrylamide levels in the pregnant woman. Finally, maternal and fetal exposure are expected to be comparable but not the same events as several factors are known to affect the amount of acrylamide that ultimately reaches the fetus (e.g., placental barrier, maternal toxic substance clearance mechanisms). Thus, such information was not recorded in this study and is not available. However, a comment about questionnaire use and alimentation is now included in our manuscript:
“The purpose of our study was to more objectively evaluate fetal exposure by studying the amniotic fluid and not based on maternal nutritional habits. Amniotic fluid measurements do not necessarily relate to maternal and placental exposure to acrylamide, as the placental barrier may work differently at this early stage of pregnancy, and the acrylamide may or may not pass to the fetus, despite the maternal exposure. Furthermore, it is extremely difficult to accurately determine maternal exposure to acrylamide with the use of questionnaires, as the same source of food may be cooked for a longer or shorter time, in different ways, etc, with these parameters affecting the acrylamide levels in the pregnant. Finally, maternal and fetal exposure are expected to be comparable but not the same occurrences as several factors are known to affect the amount of acrylamide that ultimately reaches to the fetus, such as the placental barrier explained above or the maternal toxic substance clearance mechanisms.”
- Or...In the sample in which you find acrylamide, do you have some notice about the baby? Are they in health, what about their delivery?
Thank you for your comment. The short-term outcome for all neonates included in our study was normal. Long-term outcomes were not included. Long-term neonatal outcome can be the reflection of multiple other parameters, prenatal and postnatal, such as mode of delivery, perinatal events, maternal and fetal nutritional status after delivery, etc. Certainly, it would be of great interest to undertake such a study. The following comment will be included in our manuscript:
“In our study, acrylamide was detected in only six out of 35 AGA cases. The short-term outcome for all neonates included in our study was normal. This period reflects the time of birth up until discharge of the neonate from the hospital. Longer-term consequences were not investigated in our study; they are considered to be a reflection of multiple other parameters as well. However, it would be of interest for future studies to focus on this period of neonatal and infant life.”
Round 2
Reviewer 2 Report
Comments and Suggestions for Authors
Thank you for your answers that have made me understand better the meaning of some information. The discussion that you have added is sufficient for me to permitt at your study to be accepted hoping that you would continuing to explore this field in the vision that your amniotic fluid test could become something that you could do to understand the health status of the baby.